# Varicella-Zoster Virus Prevalence among Pregnant Women: A European Epidemiological Review

**DOI:** 10.3390/life13020593

**Published:** 2023-02-20

**Authors:** Anna Bertelli, Valentina Carta, Lucia Mangeri, Arnaldo Caruso, Maria Antonia De Francesco

**Affiliations:** Institute of Microbiology, Department of Molecular and Translational Medicine, University of Brescia-ASST Spedali Civili, 25123 Brescia, Italy

**Keywords:** VZV, pregnancy, treatment, seroprevalence, vaccination

## Abstract

Europe has faced a massive spread of the varicella-zoster virus through the years. Since the introduction of an effective vaccine, complications and severe forms of chickenpox have been restricted. Nevertheless, among the population, some categories need specific care, such as pregnant women, who present one of the most fragile conditions facing this infection, both for the mother and the fetus. In this review, we highlight how the varicella-zoster virus can be dangerous during pregnancy, underlining the problem of treatment and vaccination, and collect information about the European epidemiology among this particular category of women.

## 1. Introduction

Varicella-zoster virus (VZV) is a human-specific α-herpes virus responsible for chickenpox and herpes zoster. VZV is a virus with a lipid-rich envelope acquired from cellular membranes, where viral glycoproteins are inserted. Inside the envelope, a tegument layer formed by regulatory proteins surrounds an icosahedral nucleocapsid core containing the linear double-stranded DNA genome [1,2].

VZV is responsible for two clinical pictures. Chickenpox is the primary infection, which predominantly affects children, and it is characterized by fever, vesicular, and a pruritic and painful rash, while herpes zoster is typical in adulthood and third age, and it is caused by reactivation of the VZV latency form. Particularly after the primary infection, VZV establishes latency in the sensory nerve root ganglia, and it could reactivate as a consequence of immunosuppression. VZV reactivation leads to viral replication and to the release of newly produced virus particles along the nerve pathway, causing a vesicular rash in the dermatome innerved by the affected sensory ganglion [2,3]

Since chickenpox is extremely contagious, it is one of the most common exanthematous diseases [3,4]. Although chickenpox is a mild self-limiting disease in immunocompromised children, it could be severe in adolescents and adults and particularly serious in immunocompromised subjects [4]. Furthermore, chickenpox acquired in pregnancy could affect either the mother, the fetus, or the newborn [3].

Specifically, varicella is often acquired through the inhalation of saliva droplets containing the infecting VZV and, more rarely, through direct contact with skin lesions of subjects affected by varicella or zoster [4]. Once VZV reaches the mucosal epithelial cells, the first viral replication begins, and VZV spreads to tonsils and lymphoid tissues, infecting T cells and determining a primary low-grade viremia. As a result, the virus is delivered to spleen and liver, where a second viral replication occurs. This leads to a secondary viremia, through which the virus reaches skin epidermal cells and mucous membranes, where it replicates, causing typical vesicular lesions. At this point, the infectious VZV is released through droplets from the respiratory tract [4]. Another postulated mechanism is that T cells directly transport VZV from the mucosal membrane to the skin [5].

The epidemiology of chickenpox is different between temperate and tropical climates. In temperate areas, before VZV vaccine introduction, VZV primary infection was hugely common during childhood. In fact, in North America, Europe, and Japan, more than 90% of the population contracted this disease by 15 years of age, whereas in tropical countries, only 25–85% of the child population is infected by VZV [6]. Therefore, women of childbearing age from tropical areas are more likely to be susceptible to VZV primary infection during pregnancy because they are seronegative to anti-VZV-IgG. In the UK, chickenpox incidence is estimated to be 2–3 per 1000 pregnant women; however, in the last decades, this incidence seems to be higher, probably due to increased immigration from tropical areas [3].

The aim of this review was to analyze the current epidemiology of VZV infection among pregnant women in European countries, underlining the effect of VZV infection in pregnant women, its management, and the impact of vaccination on this particular susceptible population and on newborns and children.

## 2. VZV in Pregnant Women

Varicella pneumonia is the most frequent maternal sequelae of VZV primary infection, affecting 10–20% of pregnant women with chickenpox, while encephalitis and hepatitis are rare complications [3]. Mortality and morbidity caused by this infection are higher in pregnancy than in nonpregnant women. Specifically, if the infection occurs in the third trimester, varicella pneumonia is more severe than in other pregnancy trimesters because, in this period, the uterus is so enlarged it impacts diaphragm movements [7]. Before antiviral therapy, the pregnant women mortality rate ranged between 20% and 45%, while after the introduction of acyclovir and supportive care, the mortality rate decreased to 3–14% [6,8]. During the first week after the appearance of the typical varicella rash, pneumonia symptoms include fever, dry cough, exertional dyspnea, and mild hypoxemia [6,9]. Smoking, pre-existence of respiratory illnesses, immunosuppression, and the presence of more than 100 lesions represent risk factors for the development of varicella pneumonia [3,8,9].

Fetal sequelae depend on the gestational age at which the mother is affected by chickenpox. In particular, during the first pregnancy trimester, primary infection does not increase the risk of miscarriage [3]. The occurrence of VZV infection in the first and second trimesters could lead to the development of fetal varicella syndrome (FVS), also known as congenital varicella syndrome (CVS); in particular, the highest risk (2%) is documented between the 13th and 20th gestational week [6]. The pathogenesis of FVS consists of a herpes-zoster-like reactivation in the uterus rather than a primary infection of the fetus *per se*, in which fetal immature-cell-mediated immunity is not able to counteract the virus [3,6]. After 20 weeks of gestation, FVS is very rare because the developing fetus’ immune response could be able to mount a response against the virus. Infants born after FVS are affected by neurological defects, mental disorders, unilateral limb-shortening defects with muscular hypoplasia, eye diseases, cicatricial scars in dermatomal distribution, gastrointestinal and genitourinary abnormalities, and recurrent aspiration pneumonia [1,6], and 30% of infants die within the first month of life [3,7]. Maternal herpes zoster during pregnancy does not cause FVS [1,10].

The risk of fetopathy is extremely low when mothers acquire chickenpox between the 21st and the 36th gestational week, and the prognosis is favorable. Furthermore, these children are more likely to develop an uncomplicated herpes zoster in early childhood [3].

In the last four gestational weeks, chickenpox of the mother represents a concrete risk for the newborn to develop varicella. Indeed, perinatal infection could occur in 50% of cases, and despite passively acquired antibodies from the mother, 23% of them present varicella clinical traits. Neonatal varicella could be acquired from an ascending vaginal infection through the placenta through contact with vesicular fluid at the time of delivery, or through respiratory droplets. Nowadays, the mortality rate is 7% due to antiviral therapy, neonatal intensive supportive care, and anti-VZV immunoglobulin administration, but previously, it was 30% [3,7].

Moreover, 5 days before and 2 days after delivery, maternal VZV primary infection could result in neonatal fulminant varicella because a fetus does not have the maternal anti-VZV antibodies to overcome a high viral load [3,7]. In fact, the risk of varicella in the newborn significantly diminishes when maternal varicella occurs long enough before the term in order to allow anti-VZV transfer through the placenta. However, infants born from these mothers could present vesicular lesions or develop them within 5 days after delivery, but the prognosis is good (Figure 1) [1].

## 3. Immune Response to VZV Infection and Diagnosis

### 3.1. Immunity against VZV Infection

#### 3.1.1. Cellular Immunity

Cellular-mediated immunological response to VZV is mediated by antigen-specific T lymphocytes, which are produced during primary VZV exposure and control viral replication [1,4]. Immediately, VZV infection causes the production of interferon-α (IFN-α) and -β (IFN-β) by resident skin cells, which are delivered to lesions by plasmacytoid dendritic cells [11]. Activated T cells produce interleukin-2 and interferon-γ, which increase their actions, while cytotoxic T lymphocytes act against infected cells. Moreover, natural killer (NK) cells arise early, and they eliminate infected cells. When the skin lesions are recovered, T lymphocytes appear [4,12]. Indeed, it is known that T cell immunity declines with increasing age and in immunosuppressed patients [4,13].

#### 3.1.2. Humoral Immunity

Concerning anti-VZV humoral immunity, primary VZV infection leads to the production of IgM, IgG, and IgA antibodies. Antibodies bind to VZV proteins and neutralize the virus, lysing infected cells. Clinical studies showed that anti-VZV IgG antibodies inhibit VZV infection and also in immunocompromised subjects administered with them within 72 h after exposure. In most subjects, during the incubation of primary VZV, humoral immunity appears to be limited. However, in healthy individuals, antibody presence is usually measurable within three days after the clinical symptom’s onset [14].

IgM antibodies decrease in a few months, while IgG antibodies last for years after infection [15]; indeed, primary VZV infection provides a robust humoral response that protects against possible reinfection [16]. Following primary acute infection, VZV migrates from lesions to neurons, where it persists in latency, and periodically, it could reactivate, causing zoster, whose incidence depends on the patient’s age and health conditions.

### 3.2. Laboratory Diagnosis

Laboratory diagnosis allows the detection of VZV DNA or proteins in clinical samples, such as blood, skin swabs from lesions, fluids or vesicles, cells in tissue sections of different organs, and also joint fluid, cerebrospinal fluid, or bronchial washing in the case of a disseminated infection.

#### 3.2.1. Serological Diagnosis

The 97–99% of adults who had a positive history of VZV were seropositive in their anti-VZV IgG serologic tests [6]. This is the tool implemented to establish the individuals who should be administered the varicella vaccine, and it is useful for starting antiviral therapy [1]. Indeed, VZV infection is routinely diagnosed through clinical symptoms and/or positive serological tests [6].

Regarding the serological diagnosis of primary VZV infection, it is necessary to analyze acute- and convalescent-phase serum specimens to detect IgG and IgM antibodies against VZV. However, an IgG antibodies assay is enough to determine the immune situation when the VZV history is unknown [1].

The copresence of IgM and IgG indicates recent infection or vaccination or onset of herpes zoster infection [17], while the exclusive presence of IgG attests to previous exposure to the virus and the presence of ongoing immunity. Unfortunately, the risk of obtaining false positive and false negative results in the detection of IgM and false negatives in the detection of IgG is high because these methods lack sensitivity and specificity. If the patient is symptomatic, it is advisable to repeat the test within 2–3 weeks [1,4].

Furthermore, immunofluorescence and immunoperoxidase are techniques exploited for the rapid identification of VZV viral proteins and are performed on epithelial cell samples taken from typical VZV skin lesions [18,19].

#### 3.2.2. Molecular Diagnosis

Employing a rapid, sensitive, and specific diagnosis is essential to start antiviral therapy as soon as the infection begins [1].

Some suitable sensitive and accurate diagnostic strategies are PCR methods, including loop-mediated isothermal amplification (LAMP) [20] and Southern blot or in situ hybridization [6]. Positivity to these tests does not translate into an ongoing viral infection; therefore, the clinician must also consider the patient’s condition to exclude false positivity.

#### 3.2.3. Cell Cultures

The unequivocal detection of the infectious virus is based on the use of cell cultures and on the observation of viral cytopathic effects; however, this method is not widely employed due to the long time required, from 2 to 14 days. Moreover, the cytopathic effect is similar in VZV, the herpes simplex virus (HSV), and human cytomegalovirus (CMV) infection; therefore, the isolated virus has to be further identified with virus-specific antisera [1].

## 4. Varicella Treatment

Varicella treatment can be divided into two different approaches. The pre-exposure one mainly consists of vaccination in order to prevent the clinical manifestation of varicella and to protect those subjects who are not eligible for vaccination. The second one is the postexposure approach, which can be further divided into immunoglobulin administration and antiviral drug treatment.

### 4.1. Pre-Exposure Treatment

#### 4.1.1. Varicella Vaccination

During the 1970s, the first live attenuated vaccine was developed by Takahashi et al.; this vaccine contained the Oka strain of the varicella-zoster virus passaged in guinea pig embryo fibroblasts and expanded for vaccine production in WI38 cells [1]. Starting from this vaccine, other companies developed several vaccines: Varivax, Varilrix and Zostavax are examples of live attenuated vaccines; Proquad and Priorix-Tetra are both live attenuated vaccines, but they consist of a combination of four different viruses; while others are under testing, such as the subunit vaccine Shingrix and the heat-inactivated vaccine V12 [4].

The efficacy of the currently available vaccines has been demonstrated because these vaccines induced antibody and cell-mediated responses in more than 95% of subjects enrolled in many different retrospective studies [4]. The cost–benefit ratio could be surely demonstrated by the decrease of 50–70% in clinical manifestations because of vaccination. According to previous studies, the immune response duration declines over years, with a still low level of antibodies detectable 10 years after vaccination [4,21]. Numerous countries adopted the two-vaccine-dose regimen, in which two doses are administered at two distant points in time to give longer protection against VZV. Adverse reactions to the varicella vaccine were registered in some small cohort studies, and most of them were only transient adverse effects, such as skin rash, pain, redness, and swelling at the inoculum site. The more severe adverse effects registered were those involving the central nervous system, with very rare cases of aseptic meningitis and cerebellar ataxia [4,22].

Since the varicella vaccine is an attenuated vaccine, reactivation may occur; similar to VZV, VZV Oka causes an infection, which is promptly contained by the immune system and, similar to the natural virus, it persists in the ganglia in a latent state [4,16]. This may lead to the reactivation of the virus, causing zoster even at a young age but boosting immunological memory [4,23].

Although studies have been conducted involving both children and adults, no efficacy analyses have been implemented concerning pregnant women. In addition, since the Varivax vaccine is a live attenuated vaccine, for 1 to 3 months postvaccination, the avoidance of pregnancy is advised [24].

#### 4.1.2. Immunoglobulin

Varicella-zoster immunoglobulin is a purified immunoglobulin G preparation made from human plasma containing high levels of anti-VZV antibodies [25]; it is also known as VARIZIG and represents an improved version of the previous product, VZIG, which is no longer available. This preparation is recommended for high-risk groups, including pregnant women for whom varicella vaccination is not advisable. VARIZIG administration is recommended as soon as possible following VZV exposure, ideally within 96 h to, at the latest, 10 days postexposure [25,26]. The main effect of immunoglobulin preparation consists of a reduction in varicella incidence and maternal and neonatal severity symptoms [3,27]. The recommended dose of immunoglobulin is 125 units/10 kg up to a maximum of 625 units [3,28].

### 4.2. Postexposure Treatment

#### 4.2.1. Vidarabine and IFN-Alpha

Vidarabine and IFN-α were the first antiviral agents used to treat life-threatening primary and recurrent VZV infections in immunocompromised patients [29]. Subsequently, these drugs were replaced by acyclovir, famciclovir, valacyclovir, and nucleoside analogs, which are currently licensed in VZV infection treatment [1]. Nucleoside analogs work as nucleosides during virus replication, but their structure leads to the premature termination of viral DNA polymerase activity.

#### 4.2.2. Acyclovir

Specifically, acyclovir is a synthetic nucleoside analog of guanine [24], and its mechanism exploits viral and cellular kinases in order to interrupt viral replication. Acyclovir cell specificity is determined by selective phosphorylation initially operated only by the viral thymidine kinase; this particular action allows the drug to be active only in VZV-infected cells that contain this specific enzyme. After the first phosphorylation, the monophosphate form of the compound is converted into a triphosphate form, which acts as a competitive inhibitor chain terminator of viral DNA polymerase [1].

Acyclovir has two main limits: the first one is regarding its low bioavailability, leading to frequent oral administration to achieve therapeutic levels [3,24,30]. The other one consists of the fact that acyclovir is not licensed for pregnancy administration; there are potential risks of teratogenesis with acyclovir use during the first trimester [3,24], but analysis of the registries of neonates exposed to acyclovir in utero does not show any significant risk [31,32] and, moreover, data from the pregnancy registry suggest its safety [33]. If varicella subsequently develops, pregnant women should be immediately evaluated clinically and appropriately treated with acyclovir [34]. In fact, although acyclovir does not protect the fetus from FVS or the newborn from neonatal varicella, there might be some positive effects of drug migration across placenta; this is mainly due to viral replication inhibition during maternal viremia, resulting in reduced transplacental transmission and lower fetal viremia, which is one of the main responsible for FVS, together with the gestational period of infection [3,35].

Acyclovir doses depend mainly on age, symptoms, and weight. It could be given both via oral and intravenous (IV) administration: oral acyclovir dose is 800 mg/m^2^ five times per day [3,24] for up to 7 days, and it could be reduced to thrice a day in the case of herpes zoster reactivation [3]. Intravenous administration is recommended, especially in the case of clinically relevant symptoms, from rash to severe pregnancy complications, such as pneumonia [24]: within 24–72 h from rash onset, IV acyclovir treatment should be started with a 10–15 mg/kg of body weight dose every 8 h for 5–10 days [24]. The pharmacokinetics of IV-administered acyclovir results in prolonged plasma concentrations significantly above the VZV inhibitory range [1].

Since acyclovir has been the main drug employed to treat VZV infections and due to prolonged administration at low doses, thymidine-kinase-negative VZV mutants have been selected [1,36,37,38].

#### 4.2.3. Valacyclovir

Valacyclovir is an acyclovir prodrug characterized by a longer half-life and better oral absorption [24,34] in comparison to acyclovir. It is a valine ester derivative of acyclovir, and immediately after absorption, it is converted into its parent compound [1,39]. Valacyclovir recommended dose is 1 g three times per day [24,30].

#### 4.2.4. Famciclovir

Famciclovir is a penciclovir prodrug [24], which is a guanosine nucleoside analog [1,40,41]; it is uptaken by intestinal cells, and it is completed in the liver. Similar to acyclovir, it is phosphorylated by viral thymidine kinase followed by cellular thymidine kinase, resulting in a high accumulation within VZV-infected cells [1].

#### 4.2.5. Other Drugs

Foscarnet and interferon-α (INF-α) are both licensed drugs to treat varicella, especially in those cases where the VZV strain belongs to acyclovir-resistant ones. These drugs show particular efficacy in high-risk patients. In addition, ganciclovir showed an in vitro activity against VZV, but it has not been tested in vivo yet due to its significant toxicity. Sorivudine, better known as BVaraU, is another nucleoside compound with particularly high inhibitory in vitro activity against VZV. It is well absorbed after oral administration, and it is recommended once or twice per day [1].

## 5. Management of Varicella Exposure in Pregnant Women

All women of childbearing age should undergo preconception screening for chickenpox. For those who have never developed VZV primary infection or do not know, a serological screening should be implemented in order to evaluate their immunological status against VZV. In case these women test negative for anti-VZV antibodies, they should be immunized before becoming pregnant [30]. In case of significant exposure to VZV of a pregnant woman, different pathways should be followed based on the woman’s serological status. Significant exposure to varicella means face-to-face contact or contact in the same room for 15 min or more with an affected person. Therefore, in the case of a pregnant woman, very thorough control procedures should start in order to avoid complications.

If the pregnant woman does not present a previous clinical history of chickenpox or vaccination, serology should be performed in order to evaluate her immunological status against VZV within 24–48 h from exposure. If susceptibility is confirmed, this woman should be administered with VARIZIG within 72–96 h up to 10 days after VZV exposure [3].

Ten days after exposure or in the case of a chickenpox rash development in a pregnant woman, the caregiver must be contacted immediately. First, symptomatic treatment with fever control and antipruritic drugs should be administered; subsequently, antiviral drug treatment should start with oral acyclovir 800 mg five times per day for 7 days. Antiviral therapy is mainly effective on the mother [3]; acyclovir seems to reduce fever duration and symptom severity if given to an immunocompromised subject within 24 h from rash onset [3,42]. Since there are no studies involving pregnant women, acyclovir use is not licensed but still recommended, especially in some cases; for example, when complications occur, such as respiratory symptoms, seizure, or hemorrhagic or dense rash [3,43], or if the pregnant woman is immunosuppressed, affected by chronic lung disease, or 36 weeks pregnant. If severe maternal varicella occurs, the mother should be treated with intravenous acyclovir along with supportive intensive care.

A different scenario occurs if VZV infection occurs in the last 4 weeks of pregnancy: this exposes the newborn to a significant risk of neonatal varicella [3].

In fact, during the viremia period with active vesicles, the delivery also carries a high risk of maternal hemorrhage and coagulopathy due to thrombocytopenia or hepatitis and a high risk of severe neonatal varicella. According to current recommendations, VARIZIG should be given to all infants born from mothers who have been affected by chickenpox from 7 days before to 7 days after delivery [3,44]; although VARIZIG does not prevent infection, it could reduce symptoms and disease severity. If the newborn develops varicella despite VARIZIG treatment, they should be administered intravenous acyclovir [45] (Figure 1).

## 6. European Epidemiology of VZV Infection

Before the introduction of the varicella vaccine, European serological surveillance showed an anti-VZV-IgG seroprevalence higher than 80% by the age of 10 for all countries except for Greece and 90% by the age of 15 for all countries except for Greece (86.6%) and Italy (85.3%). Anti-VZV-IgG seroprevalence data suggest that the great majority of children and adolescents seroconvert before adulthood.

Based on seroprevalence, Europe could be divided into three clusters: Belgium, Luxemburg, and The Netherlands belong to the first group, with 70% or more seroprevalence by the age of 5. In Finland, France, Germany, Iceland, Ireland, Slovenia, Spain, and Switzerland, a seroprevalence of 90% is reached by the age of 10, while in Greece, Italy, Poland, Slovakia, and the UK, a seroprevalence of 90% is reached at an older age than in the other European countries (Figure 2).

Geography does not seem to play a crucial role in varicella distribution, whereas social mixing and population structure deeply affect VZV incidence [54]. In fact, the greatest risk of being seronegative was found in women and first-generation immigrants from tropical countries, in which the population developed seropositivity after the primary infection between 25% and 85% in comparison to more than 90% in European adults [6,55].

The latest surveillance report from EUVAC considered varicella spread in Europe in 2010, and it was based on data obtained from those countries with a mandatory notification system for varicella disease.

In this report, major incidences of varicella cases were reported in four countries: Poland, the Czech Republic, Estonia, and Slovenia. In Denmark, Iceland, Luxemburg, Sweden, Switzerland, and Turkey, mandatory notification for varicella cases was not implemented [56].

### Seroprevalence in Pregnant Women

Italy was the European country with the highest number of susceptible women of childbearing age [57,58].

Pregnancy represents a particular condition in which VZV infection can be extremely dangerous. Because of this, only retrospective studies can be evaluated in order to understand seropositive rates among European pregnant women and describe anti-VZV antibody seroprevalence among the European population.

The literature data reported the values of the annual incidence of chickenpox as ranging from 1.5 to 4.6 cases per 1000 pregnant women [3,59].

In the UK, the maternal chickenpox incidence is about 1.6–4.6 per 1000 pregnant women [3,55], and a higher rate was associated with pregnant women who did not belong to a British ethnicity [60]. Particularly, the study of Talukder et al. [60] compared the risk of VZV seronegativity in pregnant women of British ethnicity to those of Bangladeshi ethnicity and, as many other studies have previously revealed, the VZV seropositive percentage was lower in immigrant people than in the European population [6,55,61].

VZV seroprevalence is higher in Northern and Western Europe than in Eastern and Southern Europe (Table 1) [62].

Several studies on VZV seroprevalence reported higher seronegativity rates in Italian and Spanish pregnant women, 10.6% and 12%, respectively, compared with other reported European rates (less than 5%) [48,50]. However, more recent data reported higher seropositivity in Spanish and French pregnant women, precisely 96.1% and 98.8%, respectively [3]. Indeed, due to the high seroprevalence rate among European pregnant women, chickenpox occurrence is infrequent in this portion of the European population [3].

A Belgian study reported 98% seropositivity among pregnant women [46], unlike Croatia, where a study including women aged 16 to 45 years showed only an 84% seropositivity rate [47]. A French population study involving pregnant women (from 19 to 43 years old) described a VZV immunization rate of 98.8% [62,63], while an Italian study concerning women aged 17 to 42 years reported a VZV seropositivity of around 81% [58]. In Scandinavian countries, the seroprevalence of VZV is higher than in other European countries, as demonstrated by a Finnish study in which anti-VZV seropositive pregnant women were 96.2% [48]. Furthermore, a Dutch study reported 100% VZV seropositivity among women aged 16 to 44 years working in nursery schools compared with a still-high percentage of 94% of other women [51].

A Norwegian study reported [52] that 98.6% of pregnant women were seropositive to VZV, while 1.2% were susceptible; of note, 28% of the latter were seroconverted during pregnancy.

Seropositivity to varicella in three different countries was evaluated using TORCH analysis: in Germany, 99.4% of women tested positive for IgG antibodies against VZV; in Poland, 98.1%; and in Turkey, 98.6% [49].

However, VZV vaccination is not recommended in pregnant women, and postpartum administration of VZIG in seronegative women is advisable for prevention in case of further pregnancies [55] and in newborns to improve neonatal outcomes [3].

## 7. Vaccination Policies

As regards VZV vaccination, in European countries, the situation is heterogeneous. In fact, this vaccine has been part of the routine vaccination program of four countries: Germany since 2004, Spain since 2006, Greece since 2006, and Latvia since 2008 [4]. Following these four countries, Luxembourg in 2009, Cyprus and Austria in 2010, and Finland in 2017 implemented VZV vaccination in the recommended list of vaccines [64]. According to a Danish study, the administration of VZIG significantly reduces the development of VZV infection and complications, such as pneumonia; furthermore, postpartum vaccination is crucial in seronegative women who develop the infection during pregnancy [55].

To date, in 17 European countries, VZV vaccination is recommended for individuals at risk and immunocompromised ones, while there are no policies for 12 nations. Italian policies regarding this vaccination changed from a regional recommendation (implemented in Sicily in 2003, Veneto in 2007, and Apulia and Tuscany in 2010) to a nationwide program in 2017 [4,65].

### Impact of Vaccination on Children and Newborns

According to the last ECDC report about varicella (2015), in Europe, reported chickenpox cases mostly involved children in the first years of life; in particular, 52–78% of infections occurred in children aged 6 years, and 89–95.9% of cases occurred before adolescence. In order to prevent varicella infection, varicella vaccination has been widely recommended for children and adults in regular or close contact with high-risk individuals. In fact, varicella vaccines are highly effective and safe. The data collected about the single-dose varicella vaccination and the two-dose vaccine showed that both regimens protected efficiently against VZV infection. However, the protection exerted by the two-dose regimen was higher than the single-dose one [53].

In particular, in Germany, which is the most experienced European country in varicella vaccination, after the introduction of universal varicella vaccination for children over 11 months of age in 2004, a reduction of 55% of varicella cases was registered for all ages, and specifically, 63% in the age group 0–4 years and 38% in the age group 5–9-years. Moreover, in 2012, VZV case reduction was 84% for all ages. Accordingly, hospitalization rates due to chickenpox also decreased after routine vaccination introduction [53].

In Italy until 2017, when varicella vaccination became mandatory for each child in order to attend primary education, vaccination against VZV with a two-dose schedule was only recommended for susceptible adolescents of 11–18 years of age and high-risk subjects (such as susceptible women of childbearing age, healthcare workers, and people living in the same household of immunocompromised individuals). However, 8 regions out of 21 introduced the universal chickenpox vaccination in childhood. Subsequently, data from Tuscany and Sicily displayed a significant decrease in varicella incidence and hospitalization rates [53].

In Spain, after the implementation of the varicella vaccination of all susceptible Spanish teenagers between 10 and 13 years of age, a rapid reduction of 77% in the chickenpox incidence rate was registered, and consequently, the hospitalization rate was halved [53].

Two different studies concern VZV seropositivity in newborns: Waaijenborg et al. described that the protection exerted by anti-VZV maternal antibodies lasted for the first 3.4 months in Dutch newborns [66], and Leuridan et al. studied the kinetics of maternal antibodies in Belgian infants. This latter study revealed that among neonates born from the 98% of mothers who tested seropositive, seropositivity waned rapidly after birth; in particular, 87% of infants presented maternal antibodies at 1 month of age, 59% of them at 3 months of age, and 0% of them at 9 months of age [46].

The latest report by EUVAC described that the highest number of VZV cases is registered among unvaccinated children below the age of 10, with complications and hospitalization rates higher in newborns younger than 1 year and, in the oldest children, older than 15 years [56].

The hospitalization rate was evaluated in different countries: in Germany, Spain, Italy, France, and Greece, it was 14.1–28/100,000 children, while in the UK and Ireland, it was 0.82/100,000 children.

Complications in hospitalization primarily occurred in the well-being of children in all the countries where VZV children cases were detected, such as France, Germany, Poland, Switzerland, Spain, Italy, and The Netherlands. The most common ones were bacterial superinfection of skin and tissue, respiratory complications, and hematological, gastrointestinal, and neurological ones [62].

## 8. Conclusions

Not only the implementation of routine chickenpox vaccination but also the monitoring of its impact on the population are crucial to assess vaccination effectiveness. An accurate surveillance program should be implemented before vaccine introduction. It should continue after the vaccine introduction, and it should include vaccine coverage, vaccine effectiveness, the occurrence of adverse events, age-specific varicella disease severity and age-specific varicella incidence, herpes zoster cases, and hospitalizations.

However, in Europe, the currently retrievable manuscripts about VZV infection and vaccine monitoring are very few and often date back many years. Therefore, there is a need to maintain VZV infection surveillance programs to understand viral circulation in the light of the VZV vaccine era and to protect people ineligible for vaccination.

## Figures and Tables

**Figure 1 life-13-00593-f001:**
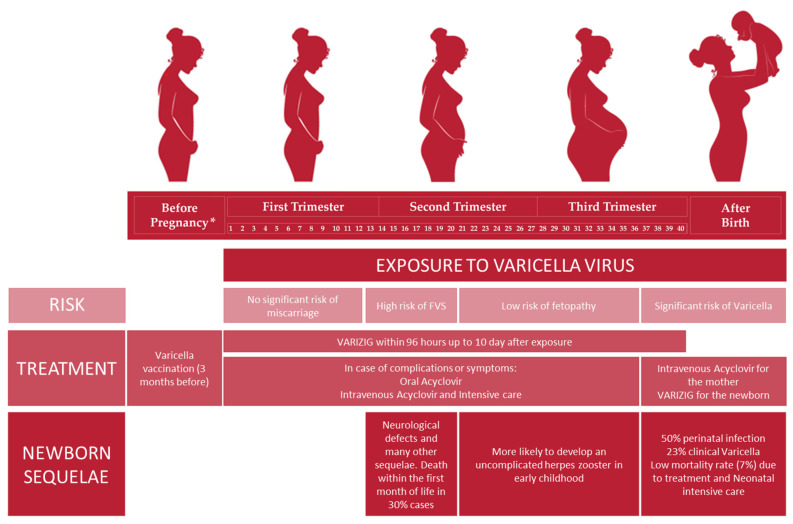
Management of VZV infection in pregnant seronegative women: risk, treatment and newborn sequelae related to the different pregnancy trimesters. * Before pregnancy: anti-VZV preconception screening is highly recommended in order to evaluate maternal serological status.

**Figure 2 life-13-00593-f002:**
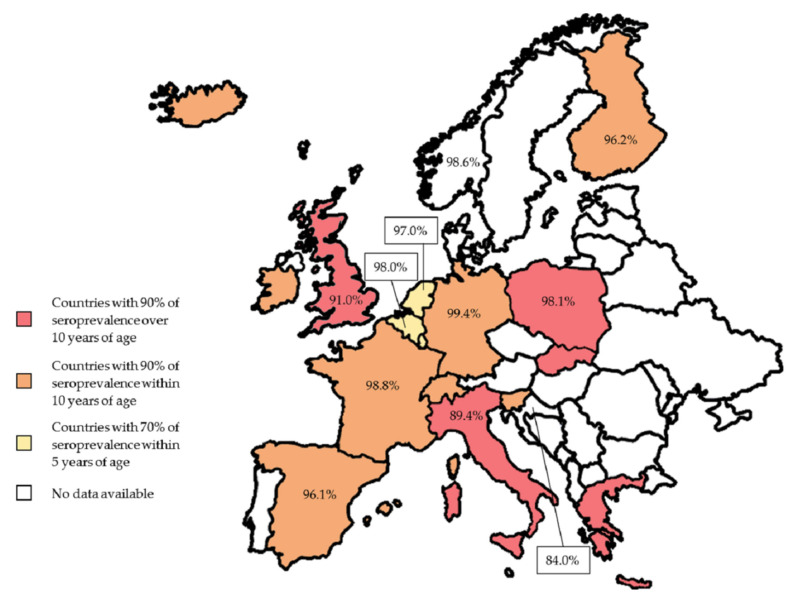
Seroprevalence in Europe with a specific focus on pregnant women. According to general seroprevalence, Europe could be divided into three clusters, shown in red, orange, and yellow, while pregnant women with seropositivity are displayed with percentage data. Data source: Belgium [46], Croatia [47], Finland [48], France [3], Germany [49], Italy [50], Netherlands [51], Norway [52], Poland [49], Spain [3], and the UK [53].

**Table 1 life-13-00593-t001:** Percentage of VZV-susceptible women in European countries.

Country	Susceptible Women (%)	Sample Collection Period	Reference
Belgium	2.0	2006–2008	[46]
Croatia	16.0	2007–2011	[47]
Finland	3.8	2000	[48]
France	1.2	2005	[3]
Germany	0.6	2006–2018	[49]
Italy	10.6	2008–2009	[50]
Netherlands	3.0	2004, 2007	[51]
Norway	1.4	1998–2009	[52]
Poland	1.9	2018–2019	[49]
Spain	3.9	2003	[3]
UK	9.0	2001–2004	[53]

## Data Availability

Not applicable.

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
