# Peer review of "Varicella-Zoster Virus Prevalence among Pregnant Women: A European Epidemiological Review"

_life, 2023, doi:10.3390/life13020593_

Round 1

Reviewer 1 Report

Authors reviewed initial VZV infection among pregnant women, especially prevalence of chickenpox and diagnosis and treatment of VZV infection. This manuscript is clearly and simply indicated. But there are some points that should be added and/or corrected.

Page 1: There is another putative mechanism that T cells transport VZV from the mucosal membrane to the skin immediately and directly (Ku CC, Zerboni L, Ito H, Graham BS, Wallace M, Arvin AM. J Exp Med 200:917-925, 2004). Authors should add the pathway.

Page 3: PCR method is sensitive and accurate diagnostic strategy. Recently, LAMP method is also effective for sensitive, accurate, and rapid diagnostic strategy (Kaneko H, Iida T, Aoki K, Ohno S, Suzutani T. J Clin Microbiol. 43:3290-6, 2005). Authors should add the method.

Page 4: “The co-presence of IgM and IgG indicates a recent infection or vaccination.” Previous manuscript was reported that VZV-specific IgG and IgM were positive in 93.9% and 12.0% of the patients with herpes zoster, respectively (Ihara H, Miyachi M, Imafuku S. J Dermatol. 45:189-193, 2018). Authors should be mentioned in this point.

Author Response

Comments and Suggestions for Authors

Authors reviewed initial VZV infection among pregnant women, especially prevalence of chickenpox and diagnosis and treatment of VZV infection. This manuscript is clearly and simply indicated. But there are some points that should be added and/or corrected.

Page 1: There is another putative mechanism that T cells transport VZV from the mucosal membrane to the skin immediately and directly (Ku CC, Zerboni L, Ito H, Graham BS, Wallace M, Arvin AM. J Exp Med 200:917-925, 2004). Authors should add the pathway.

The reference and the sentence have been added according to the reviewer’s suggestion.

Page 3: PCR method is sensitive and accurate diagnostic strategy. Recently, LAMP method is also effective for sensitive, accurate, and rapid diagnostic strategy (Kaneko H, Iida T, Aoki K, Ohno S, Suzutani T. J Clin Microbiol. 43:3290-6, 2005). Authors should add the method.

The reference and the sentence have been added according to the reviewer’s suggestion.

Page 4: “The co-presence of IgM and IgG indicates a recent infection or vaccination.” Previous manuscript was reported that VZV-specific IgG and IgM were positive in 93.9% and 12.0% of the patients with herpes zoster, respectively (Ihara H, Miyachi M, Imafuku S. J Dermatol. 45:189-193, 2018). Authors should be mentioned in this point.

The reference and the sentence have been added according to the reviewer’s suggestion

Reviewer 2 Report

In general, there are a lot of spelling and grammatical errors throughout, but overall the review is well written. I think the information provided here is a good summary for the field and would prove useful. I do, however, recommend some changes prior to publication. 

General comments: The first half of this paper relies heavily on a handful of reviews (section 1-3 rely almost exclusively on the first 4 papers). I strongly recommend adding more primary literature citations. Also, the paper in general is clearly missing citations in areas where claims have been made, examples have been highlighted below. I also think this paper would be strongly improved the addition of at least one figure and one table (table described below). A figure summarizing varicella infection, pregnancy risks, etc. would be good. Lastly, I would say a bulk of this paper is not really focused on pregnancy. Immune response section generally focuses on VZV, not VZV during pregnancy, Introduction focuses loosely on pregnancy, varicella treatment is loosely focused on pregnancy (more info on treatment in general with qualifiers about differences during pregnancy), even the epidemiology section is not strongly focused on pregnancy. While I think this information is quite useful, it is not quite as focused on pregnancy and VZV as I would have expected based on the title. 

Specific Comments

- Line 39: Citation please. Citation 4 will suffice at this place.  

- Line 47-49: I don't believe this is true. I do think most people are seropositive, but with vaccination I am pretty confident the incidence has plummeted since its usage in 1995, unless you are counting vaccination in these numbers. 

-Line 50-52: "Therefore, women in child-bearing age" This sentence is unclear, please examine and rephrase. 

- Line 63-67: citations please. I'd say at least one for pneumonia, and then a second for the antiviral. 

- Line 85-86: Please clarify this sentence "Maternal herpes zoster during pregnancy does not cause FVS". Is this trying to state that latent infection doesn't cause it?

- Line 118-120: Please clarify this sentence. Antibodies are detected within 3 days of symptom onset, but you claim the humoral immune response isn't stimulated during incubation? Not sure exactly what the point of this sentence is. 

- Lines 121-127: This paragraph feels out of sorts. Perhaps place the t-cell statements with your earlier description of t-cells and then just add the antibody section to your humoral sentence above. 

- Immunity and diagnosis section: I think this section would benefit from adding subheadings like in section 4. 

- Line 201: "and before that, it was named VZIG, which is no longer available."This is confusing, is this another type of product, or a nickname for Varizig? You make it sound as though varizig is a current treatment, but VZIG is not available?

- Line 232-233: " acyclovir does not protect the fetus from FVS or the newborn from neonatal varicella, there might be some positive effects of drug migration across placenta, due to viral replication inhibition during maternal viremia". This sentence is rather confusing. If acyclovir doesn't protect the fetus from FVS or neonatal varicella, then how can there be positive effects from drug migration across the placenta? You assert that the protection comes from reducing maternal viremia, thus drug migration across the placenta doesn't matter?
-Other drugs section: Add a discussion of complications with IFNa during pregnancy?

-Line 329: Please provide the number of women (%) that are susceptible.
- Seroprevalence section: I think this becomes too burdensome to easily digest while reading. I recommend adding a table that summarizes these findings.  
- Line 394: Citation please. 
- Line 399-401: citation please.

Author Response

Comments and Suggestions for Authors

In general, there are a lot of spelling and grammatical errors throughout, but overall the review is well written. I think the information provided here is a good summary for the field and would prove useful. I do, however, recommend some changes prior to publication. 

General comments: The first half of this paper relies heavily on a handful of reviews (section 1-3 rely almost exclusively on the first 4 papers). I strongly recommend adding more primary literature citations. Also, the paper in general is clearly missing citations in areas where claims have been made, examples have been highlighted below. I also think this paper would be strongly improved the addition of at least one figure and one table (table described below). A figure summarizing varicella infection, pregnancy risks, etc. would be good. Lastly, I would say a bulk of this paper is not really focused on pregnancy. Immune response section generally focuses on VZV, not VZV during pregnancy, Introduction focuses loosely on pregnancy, varicella treatment is loosely focused on pregnancy (more info on treatment in general with qualifiers about differences during pregnancy), even the epidemiology section is not strongly focused on pregnancy. While I think this information is quite useful, it is not quite as focused on pregnancy and VZV as I would have expected based on the title. 

According to Reviewer’s general comments, we have added more references for section 1-3. The figure has been added.

Specific Comments

- Line 39: Citation please. Citation 4 will suffice at this place.  

We thank the Reviewer for his/her comment. Citation 4 has been added

- Line 47-49: I don't believe this is true. I do think most people are seropositive, but with vaccination I am pretty confident the incidence has plummeted since its usage in 1995, unless you are counting vaccination in these numbers. 

We thank the Reviewer for his/her comment. We agree with you. In lines 51-52 of the revised version of the manuscript we have clarified that these data refer to the period before vaccine introduction.

-Line 50-52: "Therefore, women in child-bearing age" This sentence is unclear, please examine and rephrase. 

We thank the Reviewer for his/her comment. We have clarified that since in tropical areas anti-VZV IgG seroprevalence is lower than in North America, Europe and Japan, women in child-bearing age from tropical countries are more likely to be susceptible to VZV primary infection during pregnancy.

- Line 63-67: citations please. I'd say at least one for pneumonia, and then a second for the antiviral. 

We thank the Reviewer for his/her comment. In the revised version of the manuscript we have introduced the one more reference for pneumonia and another one Acyclovir.

- Line 85-86: Please clarify this sentence "Maternal herpes zoster during pregnancy does not cause FVS". Is this trying to state that latent infection doesn't cause it?

We thank the Reviewer for his/her comment. Yes, in fact the manuscript of Arvin A. M. (1996). Varicella-zoster virus. Clinical microbiology reviews, 9(3), 361–381. https://doi.org/10.1128/CMR.9.3.361 reported the following sentence “ Herpes zoster in pregnancy does not appear to result in the congenital varicella syndrome. Moreover, in this other manuscript (Pupco A, Bozzo P, Koren G. Herpes zoster during pregnancy. Can Fam Physician. 2011 Oct;57(10):1133. PMID: 21998226; PMCID: PMC3192075.) Authors agree with Arvin manuscript statement. We have added the manuscript of Pupco et al. 2011 in the reference section.

- Line 118-120: Please clarify this sentence. Antibodies are detected within 3 days of symptom onset, but you claim the humoral immune response isn't stimulated during incubation? Not sure exactly what the point of this sentence is. 

We thank the Reviewer for his/her comment, we have clarified this sentence

- Lines 121-127: This paragraph feels out of sorts. Perhaps place the t-cell statements with your earlier description of t-cells and then just add the antibody section to your humoral sentence above. 

The paragraph has been corrected according to the reviewer’s suggestion.

- Immunity and diagnosis section: I think this section would benefit from adding subheadings like in section 4. 

According to the reviewer’s suggestion, subheadings have been added in section 4.

- Line 201: "and before that, it was named VZIG, which is no longer available."This is confusing, is this another type of product, or a nickname for Varizig? You make it sound as though varizig is a current treatment, but VZIG is not available?

We thank the Reviewer for his/her comment, we have addresses this point in the revised version of the paper:

VZIG was the first immunoglobulin preparation made from individuals with a high titer of varicella immunoglobulin, in order to contrast effect of Varicella Zoster virus. Up to 2004 this product was dismissed and small amount of it where still used up to 2006. In 2006, the FDA approved a new product, still made from high titer plasma donors which happened to have varicella in a recent time. The main difference is that VZIG efficacy was declining after three days of exposure, while VARIZIG can be administered up to 10 days, still the ideal time is 92 hours.
We don’t know the difference between the two preparations. They seem to be the same product because  they are made from high titer of antibodies from plasma donors. WE supposed that it was improved in time efficacy, but still we can’t know why VZIG was dismissed.

However, we have modified the sentence, trying to explain that VARIZIG was an improvement of VIZG, which is the only reason why the product should have been changed.

- Line 232-233: " acyclovir does not protect the fetus from FVS or the newborn from neonatal varicella, there might be some positive effects of drug migration across placenta, due to viral replication inhibition during maternal viremia". This sentence is rather confusing. If acyclovir doesn't protect the fetus from FVS or neonatal varicella, then how can there be positive effects from drug migration across the placenta? You assert that the protection comes from reducing maternal viremia, thus drug migration across the placenta doesn't matter?

We thank the Reviewer for his/her comment, we have addressed this point in the revised version of the manuscript.

In FVS and neonatal varicella two factors play a crucial role: the time of infection and viremia. There is no study, for obvious reasons, of acyclovir effects on fetus, but there are records where acyclovir needed to be used on pregnant women and the fetus was then under control, in order to understand if Acyclovir would have affected it. The positive effect of drug migration across placenta is that it reduces the viremia, one of the two factors that lead to FVS and neonatal varicella. If an high viremia migrate across the placenta it can lead to FVS, while giving acyclovir it would lower it and prevent fetus from having FVS.

-Other drugs section: Add a discussion of complications with IFNa during pregnancy?

We thank the Reviewer for his/her comment,

We did not find any complications of IFNa during pregnancy.

-Line 329: Please provide the number of women (%) that are susceptible.

We thank the Reviewer for his/her comment, in the revised version of the manuscript we have added table 1 regarding this topic

- Seroprevalence section: I think this becomes too burdensome to easily digest while reading. I recommend adding a table that summarizes these findings.  

We thank the Reviewer for his/her comment, in the revised version of the manuscript we have added figure 1 regarding this topic.

- Line 394: Citation please.

We thank the Reviewer for his/her comment. Citation has been added.

- Line 399-401: citation please.

We thank the Reviewer for his/her comment. Citation  has been added